# Health Literacy among Pregnant Women in a Lifestyle Intervention Trial

**DOI:** 10.3390/ijerph19105808

**Published:** 2022-05-10

**Authors:** Farah Nawabi, Franziska Krebs, Laura Lorenz, Arim Shukri, Adrienne Alayli, Stephanie Stock

**Affiliations:** Institute for Health Economics and Clinical Epidemiology (IGKE), Faculty of Medicine and University Hospital Cologne, University of Cologne, 50935 Cologne, Germany; franziska.krebs@uk-koeln.de (F.K.); laura.lorenz@uk-koeln.de (L.L.); arim.shukri@uk-koeln.de (A.S.); adrienne.alayli@uk-koeln.de (A.A.); stephanie.stock@uk-koeln.de (S.S.)

**Keywords:** health literacy, pregnancy, lifestyle intervention, health behavior

## Abstract

Health literacy plays a crucial role during pregnancy, influencing the mother’s health behavior which in turn affects the unborn child’s health. To date, there are only few studies that report on health literacy among pregnant women or even interventions to promote health literacy. GeMuKi (acronym for “Gemeinsam Gesund: Vorsorge plus für Mutter und Kind”—Strengthening health promotion: enhanced check-up visits for mother and child) is a cluster-randomized controlled trial, aimed at improving health literacy in pregnant women by means of a lifestyle intervention in the form of brief counseling. The women in the intervention group receive counseling on lifestyle topics, such as nutrition and physical activity, during their regular prenatal check-ups. The counseling is tailored to the needs of pregnant women. Demographic data is collected at baseline using a paper-based questionnaire. Data on health literacy is collected using the Health Literacy Survey Europe with 16 items (HLS-EU-16) at baseline and the Brief Health Literacy Screener (BHLS) questionnaire at two points during the pregnancy by means of an app, which was developed specifically for the purpose of the project. The results of the study indicate that around 61.9% of the women participating in the GeMuKi study have an adequate level of health literacy at baseline. The regression analyses (general estimating equations) showed no significant effect of the GeMuKi intervention on general health literacy as measured by the BHLS (ß = 0.086, 95% CI [−0.016–0.187]). However, the intervention was significantly positively associated with pregnancy specific knowledge on lifestyle (ß = 0.089, 95% CI [0.024–0.154]). The results of this study indicate that GeMuKi was effective in improving specific pregnancy related knowledge, but did not improve general health literacy.

## 1. Introduction

Finding, understanding, appraising and applying health information—behaviors associated with having adequate health literacy—are a necessity for improving and maintaining one’s health. This becomes particularly important during pregnancy, as pregnant women have great impact on their own health and that of their unborn child. During this period, women need to possess adequate health literacy to support a healthy lifestyle during this new and challenging time period.

Pregnant women can influence the health of their unborn child through a process referred to as perinatal programming, particularly by adapting their lifestyle. More precisely, unhealthy behavior, which leads for example to excessive weight gain during pregnancy, also influences the health and growth of the unborn child. Evidence suggests that a woman’s excessive weight gain during pregnancy results in higher odds of the unborn infant developing overweightness, obesity or a chronic condition later in life [1,2,3,4]. In light of this association, it is desirable that pregnant women adhere to healthy lifestyles. Since health literacy levels are closely related to health behaviors [5,6], it is important that pregnant women are supported regarding their health literacy. Studies indicate for example, that pregnant women with inadequate health literacy are more likely to make unhealthy lifestyle choices, have more hospital stays and engage less in prenatal care [5,7,8].

There is little evidence regarding the health literacy levels of pregnant women in Germany. Studies from a recent international systematic review reported mixed findings, with some studies indicating that health literacy among pregnant women was adequate, while others indicated it was inadequate [9]. Nonetheless, researchers are in agreement that adequate health literacy during pregnancy facilitates a healthy lifestyle [6], informed decision making, and knowledge concerning prenatal tests [10,11].

In order to achieve adequate health literacy, interventions with a particular focus on strengthening health literacy are needed. Previous research indicates that health literacy sensitive interventions, by enhancing knowledge on the matter at hand, can be effective in improving health literacy, and are also beneficial in the promotion of a healthy lifestyle [12], regardless of the educational level. To date, there exist only a small number of interventions that focus specifically on the improvement of health literacy in pregnant women [13]. The GeMuKi (acronym for “Gemeinsam Gesund: Vorsorge plus für Mutter und Kind”—Strengthening health promotion: enhanced check-up visits for mother and child) intervention consists of brief lifestyle counseling sessions with a focus on pregnancy related topics. The counseling sessions are implemented into routine prenatal check-ups [14]. The intervention is aimed at strengthening health literacy among pregnant women. The purpose of this study is to assess the status quo with regard to health literacy levels of pregnant women enrolled in the GeMuKi trial, and to evaluate whether the GeMuKi intervention improved health literacy levels [15].

## 2. Materials and Methods

### 2.1. Study Design

Health literacy was assessed within the GeMuKi trial, a lifestyle intervention implemented between October 2017 and March 2022. The cluster-randomized controlled trial took place in ten regions in the state of Baden-Wuerttemberg, Germany. Five of the regions provided the intervention while the other five provided regular care. The intervention, which took the form of brief counseling sessions, took place over the course of up to six check-ups during pregnancy. It was provided by gynecologists and midwives, in case the women opted for midwifery care. The primary outcome was the prevention of excessive gestational weight gain during pregnancy. The secondary outcomes were the improvement of maternal and infant health outcomes, and improved health literacy. This paper focuses particularly on health literacy; the publication of results of the primary outcome and other secondary outcomes are in progress. A detailed description of the general design of the GeMuKi Project can be found elsewhere [14,15].

### 2.2. The GeMuKi Intervention

The GeMuKi intervention consists of brief lifestyle counseling sessions implemented as part of routine check-ups during pregnancy. Participants in the intervention group received additional counseling as part of their antenatal care, while the control group received care as usual. Participants of both groups filled in one paper-based questionnaire at baseline, and further questionnaires using an app developed for the purpose of the study.

### 2.3. Health Literacy Strengthening Components

The GeMuKi intervention aims to strengthen the health literacy of pregnant women by actively involving them in brief lifestyle counseling. The women decided for themselves which lifestyle topic they would like to receive counseling on, thus promoting participation that is key to improving health literacy [16]. The topics were recommended by the ‘Healthy Start—Young Family Network’ (Netzwerk Gesund ins Leben), which is a national network that aims to promote a healthy lifestyle during pregnancy. Its recommendations are based on systematic reviews [17]. Prior to the start of the intervention, the healthcare providers received training on communicating the key messages from the recommendations using motivation interviewing (MI) techniques [18]. Since the primary outcome of the study was gestational weight gain, this was the main focus of the training content. MI is based on the notion that individuals will change their behavior autonomously, which is considered a health literacy skill [19]. During the counseling sessions, healthcare providers listened to the information needs of the women and communicated using open-ended questions in order to trigger behavior change. This is in line with the ‘German Action Plan Health Literacy’, which states that healthcare providers should communicate sensitively according to the health literacy status of the patient in order to respond appropriately and strengthen the patient’s health literacy [16]. At the end of the counseling session, the healthcare provider and the patient jointly came up with SMART (Specific, Measurable, Achievable, Reasonable, Time-Bound) lifestyle goals to be accomplished by the next counseling session. This ensured that the SMART goals were individual and tailored to the particular health literacy levels of the women in question.

In addition to the counseling, the intervention also made use of digital components to promote health literacy as recommended by the action plan [16]. The GeMuKi-App provided the women participating in the intervention with a collection of hyperlinks related to health information on pregnancy. The app is easy to use, making it accessible for women with different health literacy and education levels. The participants also filled in questionnaires on health literacy using the app. The healthcare providers were provided with a digital interface, the GeMuKi-Assist counseling tool. The tool included supporting questions on each lifestyle topic for the healthcare providers to refer back to during counseling. In order to ensure that they were aligned with the health literacy levels of the respective women, these supporting questions were based on the principles of MI, i.e., they were open-ended questions that would trigger communication on the part of the woman in question. The healthcare providers entered the jointly agreed SMART goals into GeMuKi-Assist, after which they were displayed in the woman’s app in the form of push notifications.

### 2.4. Sample and Recruitment

Gynecologists participating in the GeMuKi trial recruited eligible pregnant women based on predefined inclusion and exclusion criteria. The women were deemed eligible if they were ≥18 years old, <12 weeks of gestation, carried statutory health insurance and possessed proficient language skills in German. The sample size was calculated based on the primary outcome of the GeMuKi study, for which a sample size of 1860 participants was required [14].

### 2.5. Ethical Approval

GeMuKi was approved by the ethics committee of the University Hospital of Cologne (ID: 18-163) and the State Chamber of Physicians in Baden-Wuerttemberg (ID: B-F-2018-100). Inference to study participants is not possible since the collected data is pseudonymized in accordance with the EU General Data Protection Regulation (GDPR). Written informed consent was obtained prior to participation in the study. The participants were reassured that they were free to withdraw from the study at any time without consequences.

### 2.6. Data Collection

The data for the analysis in this paper were derived from two sources. The women filled in a paper-based questionnaire at baseline (before the 12th week of gestation) in order to provide demographic variables. Data regarding health literacy and knowledge of pregnancy related lifestyle topics were collected using questionnaires via the app.

### 2.7. Health Literacy Assessment

Health literacy was assessed using two questionnaires. The German version of the HLS-EU-16 was utilized at baseline (t0) to provide a detailed picture of the health literacy level distribution of the pregnant women in our sample compared to the general population.

The HLS-EU-16 is based on the health literacy definition of Sørensen et al. (2012), which is based on a broad conceptualization of health literacy, including functional and critical health literacy.

The participants were asked to answer questions on a 5-point Likert scale (‘Very difficult’–‘Very easy’; ‘I don’t know’). Since paper-based questionnaires also allow individuals to skip questions, we added the option ‘I do not want to answer this question’ to the app-based survey.

In order to observe changes in health literacy levels after the intervention, we utilized a modified version of the Brief Health Literacy Screener (BHLS). This likewise allowed participants to answer questions on a 5-point Likert scale ranging from ‘Never’ to ‘Always’. We also added ‘I do not want answer this question’ as an option. This questionnaire was used both at baseline (t0) and at the end of the pregnancy (t1).

Since the HLS-EU-16 and BHLS are both subjective health literacy measures which gauge general health literacy skills, we developed a knowledge-based questionnaire to provide objective estimates of pregnancy related health literacy [20,21]. The questionnaire was developed based on the topics from the national recommendations discussed during counseling. They cover: weight development during pregnancy, portion size, nutrition, alcohol consumption, smoking, physical activity, water intake and breast feeding. The questionnaire was applied at two time points, namely t0 and t1. The answering scale was ‘Yes/No/I don’t know’. A detailed description of this questionnaire can be found in the Appendix A while the baseline results can be found elsewhere [20].

### 2.8. Data Analysis

Plausibility checks for the data were performed throughout data collection and prior to analysis. Descriptive statistics were used to analyze participant characteristics. Age, parity (nullipara), net income, migration background and educational level were used as independent variables in multiple regression analysis. Age was defined as a continuous variable. Nullipara and migration background were defined as dichotomous variables (Yes/No). Net income was calculated as a continuous variable. Education level was used as an ordinal variable. Percentages are provided for categorical variables and means for continuous variables.

The HLS-EU-16 was analyzed based on official recommendations [22]: First, the individual items of the HLS-EU-16 were binarized by collapsing the two outer answer categories (“Very easy”/“Fairly easy” = 1; “Fairly difficult”/“Very difficult” = 0). Once this had been done, a sum score was calculated using the 16 binarized items. Respondents with more than two missing answers were excluded from the sum score calculation. The sum score was categorized into three health literacy levels: Adequate (score 13–16), Problematic (score 9–12) and Inadequate (score 1–8). This allowed for a display of the status quo with regard to health literacy level among pregnant women in accordance with prior population-based surveys in Germany and international studies [23,24,25].

The BHLS is a three-item questionnaire designed to assess health literacy status by asking about confidence using health-related forms [26]. Answers can be given on a 5-point Likert scale ranging from ‘Never’ to ‘Always’. Scores can range from 5 to 15 points. Means are provided as a classification of health literacy.

To answer the question of whether the GeMuKi intervention improved the general health literacy of pregnant women, multiple regression models using general estimation equations (GEE) were used to account for the cluster structure of the study. This GEE included the deviation (Δ) of the BHLS sum score between the two time points (BHLS sum score at t1—BHLS sum score at t0) as a continuous dependent variable and group (intervention group = 1, control group = 0) as an independent variable adjusted for age, nullipara, income, migration background and education level (covariates). To answer the question of whether the GeMuKi intervention improved specific pregnancy related and knowledge-based health literacy, a second GEE model was tested using the same independent variables and covariates and the deviation of the knowledge questionnaire sum score between the two time points (knowledge sum score at t1—knowledge sum score at t0) as the dependent variable. The sum score was calculated by adding the number of correct answers for every single question [20]. Since health literacy was a secondary outcome of the cluster randomized controlled trial, no imputations were conducted. A *p*-value of <0.05 indicated statistical significance.

All the analyses were conducted using IBM^®^ SPSS^®^ Statistics for Windows, Version 28.0 (Chicago, IL, USA).

## 3. Results

The mean age of the study participants was 31; half of the study population did not have any children at the time of participation (50%). The mean household net income was EUR 4293 per month. More than half of the pregnant women had a university degree (55.1%) and 22.7% came from a migration background (Table 1).

Health literacy levels in the GeMuKi study population were adequate, with 66.5% (n = 908) of the sample possessing adequate health literacy. Around one third of the women who participated in the study possessed inadequate (5.3%; n = 73) or problematic (28.1%; n = 384) health literacy (Figure 1). Descriptive analysis of the BHLS revealed that participants had a mean score of 13.56 (n = 1373) at t0, and 13.54 at t1 (n = 1175).

Multivariable regression analysis using the BHLS as the dependent variable did not show any intervention effects on the improvement of health literacy (ß = 0.086, 95% CI [−0.016–0.187]) (Table 2). No significant association was observed for the covariates age (ß = 0.000, 95% CI [−0.035–0.036]), migration (ß = −0.127, 95% CI [−0.366–0.112]), income (ß = 5.676, 95% CI [−2.583–3.718]), education (ß = −0.034, 95% CI [−0.089–0.021]) or parity (ß = 0.061, 95% CI [−0.072–0.194]).

Table 3 displays the GEE using Δ knowledge as the dependent variable. It was possible to observe a significant positive effect of the intervention on knowledge in the intervention group (ß = 0.089, 95% CI [0.024–0.154]). The only other significant association was seen with parity indicating that knowledge gain was predicted by giving birth for the first time (ß = 0.160, 95% CI [0.059–0.261]).

## 4. Discussion

This study is the first to our knowledge that provides data on the status quo for health literacy among pregnant women in Germany and assesses whether a lifestyle intervention during pregnancy improved general or pregnancy related health literacy levels.

The first point to note is that the analysis of the HLS-EU-16 indicates that about 33% of the pregnant women had inadequate or problematic health literacy, and more than two thirds of the pregnant women had sufficient health literacy right at the initiation of the GeMuKi trial. This is above average when compared to national and international studies on health literacy. A repeated representative study using the HLS-EU-16 in Germany from 2021 demonstrated that 59% of the population have problematic or inadequate health literacy [24], which is a decline of about five percentage points to 54% compared to 2016 [23]. The same holds true for data on the female population from the 2021 study, in which around 57% of the study participants possessed inadequate health literacy.

The comparatively high baseline health literacy levels among the trial participants were not surprising considering that the study population was highly educated, which is strongly associated with health literacy [19], in comparison to the general population. Several explanations as to why our population was highly educated may apply in this case. The region in which the study took place ranks highly in national comparisons of educational achievements, making it likely that more educated women would participate in the study. It is possible that some women with migration backgrounds were not recruited by healthcare providers for the study due to insufficient language skills, which rules out one important vulnerable group. The inclusion criteria for participation in the study were set broadly; however, the choice of which women to include in the study was left to the gynecologists, which might have led to selection bias. This might also provide an explanation as to why we had so few women with a migration background in the study sample compared to the general population. Such women might have been excluded due to language barriers or because healthcare providers did not perceive them as eligible.

Multiple regression analysis using GEE did not show significant results regarding the question of whether this lifestyle intervention improved general health literacy. There are several explanations for why the intervention was not effective. Firstly, this again may be a result of the highly educated nature of the study population and the high initial levels of health literacy in both the intervention and control groups. Secondly, since it only contains three items, the BHLS does not offer a broad picture of health literacy. This could have been avoided by applying the HLS-EU-16 with 16 items at t1 as well as t0. For future studies we therefore highly recommend using a more detailed health literacy instrument. Thirdly, the intervention was not geared to general health literacy, which is measured using HLS-EU-16 and the BHLS. Therefore, the utilized instruments may have not been suitable. To depict pregnancy specific health literacy, we have developed a knowledge-based questionnaire. Fourthly, regarding the training the healthcare providers received, it can be argued that they might not have been educated well enough in the use of MI techniques which were supposed to be health literacy sensitive. It should be added that the content of the training mainly focused on the primary outcome of the intervention, which was gestational weight gain, rather than health literacy. The implementation of the counseling was not monitored, so we cannot assume that all the steps were conducted as taught in the training sessions. Lastly, we might have not met the needs of women with inadequate health literacy in our population. This can indicate that the intervention was not appropriate for that proportion of women.

According to the results of the GEE, the intervention was effective in improving the knowledge of pregnancy related lifestyles of women in the intervention group. The assessment of knowledge change in our study may have been successful as we developed a pregnancy specific health literacy instrument. The contents of the questionnaire were based on topics that the women received counseling on. This again speaks for the utilization of appropriate instruments for future studies, or that interventions are built based on the theoretical construct of the questionnaire, such as the HLS-EU-16.

The fact that the counseling helped pregnant women gain knowledge on pregnancy specific lifestyle topics can be seen as an argument in its favor. Scholars in the German healthcare setting support the provision of health information in the antenatal setting through gynecologists to improve pregnancy related lifestyle knowledge, since it has the potential to reach women of different socio-demographic status [27]. Counseling becomes particularly important regarding significant results for women that are going to be first-time mothers, as they are new to the experience of being pregnant and potentially need counseling on lifestyle during pregnancy.

Studies indicate that educational interventions to improve knowledge on pregnancy specific topics are effective; however, the transition from knowledge to behavior still requires research [28,29]. Small scaled interventions, on the other hand, already show promising results in improving knowledge of physical activity and nutrition, and hence improving behavior [30]. Similarly, interventions (also in the form of counseling) to reduce the risk of gestational weight gain were proven effective [31,32].

## 5. Conclusions

This study indicates that most women participating in a lifestyle intervention trial in Germany possessed adequate health literacy in our study population. Nevertheless, pregnant women with inadequate health literacy, who still make up about one third of the study population, should not be neglected, due to the effects that limited health literacy can have on the woman’s health and that of their unborn child. The intervention was not able to improve general health literacy; this may be due to several determinants. The women included in our study possess both high levels of education and adequate health literacy. The needs of women with inadequate health literacy might not have been met, which might stem from the training that the healthcare providers received which did not consider different health literacy-based subgroups of women. The main focus of the training was on the primary outcome of the study, the prevention of excessive gestational weight gain. Nevertheless, the study had a significant positive effect on knowledge levels, which provides strong support for providing additional lifestyle counseling during pregnancy, especially for first-time mothers.

Future interventions might benefit from a comprehensive approach to measuring health literacy throughout the study period, rather than using short screeners. Additionally, approved communication methods for increasing health literacy need to be an inherent part of counseling.

## Figures and Tables

**Figure 1 ijerph-19-05808-f001:**
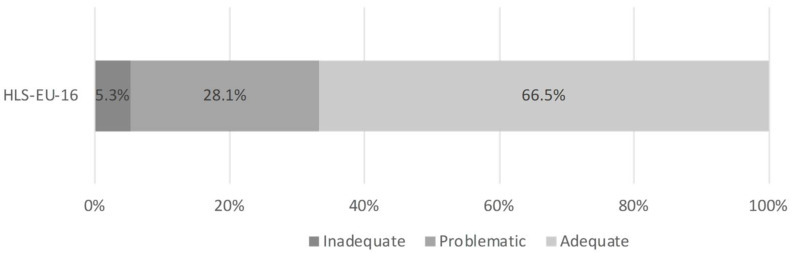
Results of the HLS-EU-16 (n = 1365).

**Table 1 ijerph-19-05808-t001:** Demographic variables of study participants.

	Total n (%)	Intervention Group n (%)	Control Group n (%)
Age in years, mean (SD)	31.3 (4.3)	31.3 (4.2)	31.2 (4.4)
Nullipara	711/1422 (50.0)	366 (47.9)	345 (52.4)
Migrant	329/1447 (22.7)	197 (25.4)	132 (19.7)
Income in euros, mean (SD)	4293 (1663)	4304 (1706)	4281 (1613)
Education level			
Primary	2/1404 (0.1)	0.0 (0)	2 (0.3)
Lower secondary	39/1404 (2.8)	20 (2.6)	19 (2.9)
Upper secondary	590/1404 (42.0)	331 (43.6)	564 (40.2)
University degree	773/1404 (55.1)	408 (53.8)	365 (56.6)

**Table 2 ijerph-19-05808-t002:** Multiple regression analysis using GEE (dependent variable: ΔBHLS, n = 1010).

Independent Variable	ß *	SE *	*p*-Value *	95% CI *
Group	0.086	0.051	0.099	−0.016–0.187
Age	0.000	0.018	0.979	−0.035–0.036
Migrant	−0.127	0.121	0.299	−0.366–0.112
Income	5.676	1.607	0.724	−2.583–3.718
Education	−0.034	0.028	0.224	−0.089–0.021
Nullipara	0.061	0.067	0.372	−0.072–0.194

Note: ß = regression coefficient; SE = standard error; * all values are adjusted.

**Table 3 ijerph-19-05808-t003:** Multiple regression analysis using GEE (dependent variable: Δ knowledge, n = 1016).

Independent Variable	ß *	SE *	*p*-Value *	95% CI *
Group	0.089	0.033	0.007	0.024–0.154
Age	−0.003	0.006	0.697	−0.016–0.010
Migrant	−0.011	0.050	0.835	−0.110–0.089
Income	−1.922	2.187	0.380	−6.210–2.367
Education	0.008	0.031	0.805	−0.054–0.070
Nullipara	0.160	0.051	0.002	0.059–0.261

Note: ß = regression coefficient; SE = standard error; * all values are adjusted.

## Data Availability

The datasets used and/or analyzed during this study are available from the corresponding author on reasonable request.

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
