# Peer review of "Health Literacy among Pregnant Women in a Lifestyle Intervention Trial"

_ijerph, 2022, doi:10.3390/ijerph19105808_

Round 1

Reviewer 1 Report

Very interesting study - you did a fine job noting the lack of HL improvement and describing potential reasons why. 

Line 38-39.  Is weight gain the only unhealthy behavior?  Please re-write the sentence to indicate that this is an example of an unhealth behavior.

Lines 57-58:  the study referenced in (12) is not about pregnant women.  The sentence after (12) should not be specific to pregnant women without a citation.

Lines 76-77: should this be obstetricians instead of gynecologists?

Lines 102-104:  behavior change is based on critical health literacy.  I think it would be helpful to describe how that is different from fundamental health literacy.

Lines 269-270.  Immigration status is not the only thing that makes a person vulnerable. 

Author Response

Dear Editors, Dear Reviewers,

Thank you for considering our manuscript for publication and reviewing it thoroughly. We appreciate the constructive feedback and hope we were able to address all comments and suggestions in sufficient detail. Below we provide our answers and the corresponding changes in the manuscript for each comment. For easier referencing, we numbered them in consecutive order.

Reviewer 1

  1. Line 38-39.  Is weight gain the only unhealthy behavior?  Please re-write the sentence to indicate that this is an example of an unhealth behavior.
    • Thank you very much for this suggestion, which is true since weight gain is only one example of what unhealthy behavior might result into. We have added this in line 39.
  2. Lines 57-58:  the study referenced in (12) is not about pregnant women.  The sentence after (12) should not be specific to pregnant women without a citation.
    • Thank you for making us aware of this inconsistency. We have corrected this in line 57.
  3. Lines 76-77: should this be obstetricians instead of gynecologists?
    • In this study, gynecologists is the correct term and profession that we refer to, because they were the professional group that was recruited and included in the GeMuKi study. We acknowledge that there is considerable overlap between gynecologists and obstetricians in Germany, however. In the German health care system gynecologists, who conduct pregnancy check-ups, are working in ambulatory settings. They often are the first line of contact for women before they get referred to more specialized hospital care and usually serve women in childbearing age and all other age groups.
  4. Lines 102-104:  behavior change is based on critical health literacy.  I think it would be helpful to describe how that is different from fundamental health literacy.
    • In this study we did not distinguish between critical and fundamental health literacy. Zarcadoolas et al. (2005) distinguish between fundamental and critical health literacy, which Sörensen and colleagues have considered in their review and forming their definition of health literacy (Sörensen et al., 2010). Since we base our theories on the definition of Sörensen et al. who have considered these two forms in their definition, a differentiation seems not necessary. To clarify this point, we have added a sentence explaining the health literacy definition of Sörensen (lines 162-164).
  5. Lines 269-270.  Immigration status is not the only thing that makes a person vulnerable.
    • Thank you for this feedback, we have rephrased the sentence (lines 272-273).

Reviewer 2 Report

Reviewer

Thank you for the opportunity to reviewing this manuscript. As a women’s health professional, the topic is interesting, and it introduces an innovative aspect. The authors state that the purpose of this study is to assess the status quo with regard to health literacy levels of pregnant women enrolled in the GeMuKi trial and to evaluate whether the GeMuKi intervention improved health literacy levels.

The methods are appropriate, accurate, and objectivity for the experiments, and improves the understanding of the reader. Overall, the article is well written and has an excellent structure to validate the proposed objectives. But, for the manuscript to be considered, some points need to be improved.

Line 16-17: Specify more clearly about the “counseling on lifestyle related topics” with brief examples.

Line 19: The full name of a term should be indicated when the term firstly used in the manuscript. Please, give the full name of HLS-EU-16 and BHLS.

Line 55: It is advisable to specify more clearly about “health literacy sensitive interventions”

Line 77-78,101: It is advisable to replace the word “the primary outcome” to “the primary objective” or “the primary aim”.

Line 132: It is advisable to revise the word “<12th week of gestation”. We don’t use this term in normal. Instead, we use “12 weeks”.

Line 159: It is advisable to replace the word “due to” to “after”.

Line 257: Reference number [23,24] is about national studies? Please, provide relevant references regarding “international studies”

I hope my comments will be useful in the process of revising this manuscript.

Author Response

Dear Editors, Dear Reviewers,

Thank you for considering our manuscript for publication and reviewing it thoroughly. We appreciate the constructive feedback and hope we were able to address all comments and suggestions in sufficient detail. Below we provide our answers and the corresponding changes in the manuscript for each comment. For easier referencing, we numbered them in consecutive order.

Reviewer 2

  1. Line 16-17: Specify more clearly about the “counseling on lifestyle related topics” with brief examples.
    • Thank you for this suggestion, which we have added in line 17.
  2. Line 19: The full name of a term should be indicated when the term firstly used in the manuscript. Please, give the full name of HLS-EU-16 and BHLS.
    • We have added this in lines 19 and 20.
  3. Line 55: It is advisable to specify more clearly about “health literacy sensitive interventions”
    • This is a useful suggestion which we have added in lines 59.
  4. Line 77-78,101: It is advisable to replace the word “the primary outcome” to “the primary objective” or “the primary aim”.
    • Thank you for this suggestion. Because we have already used the term ‘primary outcome’ in all other papers (including the study protocols), we would like to stick to this term for consistency.
  5. Line 132: It is advisable to revise the word “<12th week of gestation”. We don’t use this term in normal. Instead, we use “12 weeks”.
    • We have adjusted this as suggested.
  6. Line 159: It is advisable to replace the word “due to” to “after”.
    • We have corrected this.
  7. Line 257: Reference number [23,24] is about national studies? Please, provide relevant references regarding “international studies”
    • Thank you for this suggestion, which we have added Sörensen et al (2015) (reference 25), as this is the most relevant interventional study conducted in several European countries.

Reviewer 3 Report

The manuscript submitted by Nawabi et al. brings important results of a multicenter study. The study is well designed, the parameters of the study are defined appropriately and the aim of the study has a high impact in society. Even the questions do not go in depth, the results will represent a baseline for (potential) future studies and for comparison between (possible) other countries. 

The project, and therefore the manuscript, highlights the importance of health literacy and the mode this approach can improve health parameters and outcomes.

I would recommend including some graphics in the results section.

Author Response

Dear Editors, Dear Reviewers,

Thank you for considering our manuscript for publication and reviewing it thoroughly. We appreciate the constructive feedback and hope we were able to address all comments and suggestions in sufficient detail. Below we provide our answers and the corresponding changes in the manuscript for each comment. For easier referencing, we numbered them in consecutive order.

Reviewer 3

  1. I would recommend including some graphics in the results section.
    • Thank you for this suggestion. We displayed the results from the HLS-EU as a figure, to make the health literacy distribution more vivid.

Round 2

Reviewer 2 Report

Accept